# Total Flavones of *Rhododendron* Protect Against Ischemic Cerebral Injury by Regulating the Phosphorylation of the RhoA-ROCK_2_ Pathway via Endothelial-Derived H_2_S

**DOI:** 10.3390/cimb47070513

**Published:** 2025-07-03

**Authors:** Xiaoqing Sun, Xingyu Zhang, Yuwen Li, Jiyue Wen, Zhiwu Chen, Shuo Chen

**Affiliations:** 1School of Pharmaceutical Sciences, Anhui Medical University, Hefei 230032, China; 15655118917@163.com (X.S.); 18019068741@163.com (X.Z.); 18326628591@163.com (Y.L.); wenjiyue139@aliyun.com (J.W.); 2The Experimental Research Center, Anhui University of Chinese Medicine, Hefei 230038, China

**Keywords:** total flavones of *Rhododendron*, H_2_S, ischemic cerebral injury, RhoA at Ser188, ROCK_2_ at Thr436

## Abstract

This study aims to investigate the mechanism by which the total flavones of *Rhododendron* (TFR) protect against cerebral ischemic injury through the endothelial-derived H_2_S-mediated regulation of RhoA phosphorylation at the Ser188 and Rho kinase 2 (ROCK_2_) phosphorylation at Thr436. For experimental design, mouse or rat cerebrovascular endothelial cells (ECs) were cultured with or without neurons and subjected to hypoxia/reoxygenation (H/R) injury. The vasodilation of the cerebral basilar artery was assessed. Cerebral ischemia/reperfusion (I/R) injury was induced in mice by bilateral carotid artery ligation, followed by Morris water maze and open field behavioral assessments. The protein levels of cystathionine-γ-lyase (CSE), 3-mercaptopyruvate sulfurtransferase (3-MST), RhoA, ROCK_2_, p-RhoA (RhoA phosphorylated at Ser188), and p-ROCK_2_ (ROCK_2_ phosphorylated at Thr436) were quantified. Additionally, the activities of RhoA and ROCK_2_ were measured. Notably, TFR significantly inhibited H/R-induced H_2_S reduction and suppressed the increased expression and activity of RhoA and ROCK_2_ in ECs, effects attenuated by CSE or 3-MST knockout. Moreover, TFR-mediated cerebrovascular dilation was reduced by RhoA or ROCK_2_ inhibitors, while the protective effect of TFR against cerebral I/R injury in mice was markedly attenuated by the heterozygous knockout of ROCK_2_. In the ECs-co-cultured neurons, the inhibition of TFR on H/R-induced neuronal injury and decrease in H_2_S level in the co-culture was attenuated by the knockout of CSE or 3-MST in the ECs. TFR notably inhibited the H/R-induced upregulation of neuronal RhoA, ROCK_2_, and p-ROCK_2_ protein levels, as well as the activities of RhoA and ROCK_2_, while reversing the decrease in p-RhoA. However, the knockout of CSE or 3-MST in the ECs significantly attenuated the inhibition of TFR on these increases. Furthermore, 3-MST knockout in ECs attenuated the TFR-mediated suppression of p-RhoA reduction. Additionally, CSE or 3-MST knockout in ECs exacerbated H/R-induced neuronal injury, reduced H_2_S level in the co-culture system, and increased RhoA activity and ROCK_2_ expression in neurons. In summary, TFR protected against ischemic cerebral injury by endothelial-derived H_2_S promoting the phosphorylation of RhoA at Ser188 but inhibited the phosphorylation of ROCK_2_ at Thr436 to inhibit the RhoA-ROCK_2_ pathway in neurons.

## 1. Introduction

Ischemic brain injury is a prevalent cause of mortality and disability worldwide, significantly impacting human health and life quality [1]. Chinese herbal medicine has a long history of use in the prevention and treatment of ischemic cerebrovascular diseases, with advantages in multi-target and multi-component comprehensive therapeutic effects [2,3]. Extracting active ingredients from Chinese herbal medicines has been an important approach for discovering new drugs in the treatment of ischemic brain injury [4]. The total flavones of *Rhododendron* (TFR), an effective component of flavonoids extracted from *Rhododendron*, mainly consists of quercetin, hyperoside, rutin, and other flavonoids, and is a significant medicinal resource [5]. In multiple cerebral ischemia/reperfusion (I/R) injury models [6,7,8], TFR exerts significant protective effects against ischemic brain injury by enhancing hydrogen sulfide (H_2_S) production and inducing cerebral vasodilation. However, its exact mechanism requires further investigation, particularly on which downstream signaling pathway of H_2_S involves the protective effect of TFR.

The production of endogenous H_2_S occurs through the direct enzymatic desulfhydration of cysteine, which is catalyzed by cystathionine-γ-lyase (CSE) and cystathionine-β-synthase (CBS). Additionally, an indirect desulfhydration process is facilitated by 3-mercaptopyruvate sulfurtransferase (3-MST) in the presence of reductants within mitochondria [9]. CBS is predominantly located in nerve cells [10], while CSE serves as the primary H_2_S-produing enzyme in blood vessels and mainly expresses in vascular endothelium [11]. However, 3-MST can exist in various cell types, including vascular endothelial cells (ECs) [12]. Therefore, the generation of endothelial-derived H_2_S is mainly catalyzed by CSE and 3-MST.

Previous studies conducted by our research group indicate that the inhibition of the RhoA-Rho kinase (ROCK) pathway may play a critical role in H_2_S-induced cerebral vasodilation and neuroprotective effects [13,14,15]. The RhoA-ROCK signaling pathway serves as a key target for cardiovascular and cerebrovascular diseases [16,17], is extensively distributed throughout the nervous system, and is intricately associated with neuronal physiological functions as well as the repair and regeneration of nerve axons [18,19,20]. ROCK has two subtypes, namely ROCK_1_ and ROCK_2_, with ROCK_2_ primarily being distributed in the brain and blood vessels [21,22]. Therefore, the RhoA-ROCK_2_ signaling pathway may serve as the primary mechanism through which H_2_S exerts its effects on regulating cerebral vascular function and providing protection against ischemic brain injury.

Protein phosphorylation, catalyzed by kinases, transfers phosphate groups from ATP to specific residues (serine, threonine, and tyrosine) on target proteins, whereas dephosphorylation, mediated by phosphatases, removes these groups [23]. These processes are essential mechanisms for regulating protein activity and function.

The phosphorylation of RhoA at Ser188 inhibits its activation and translocation from the cytoplasm to the cell membrane [24]. Additionally, the phosphorylation of ROCK_2_ at specific sites plays a crucial role in the regulation of its activity [25,26]. Our recent research revealed that exogenous H_2_S can inhibit the phosphorylation of ROCK_2_ at Thr436 in neurons, thereby playing a protective role in neuronal hypoxia/reoxygenation (H/R) damage [27]. TFR promotes H_2_S production to exert protective effects against cerebral I/R injury [7]; however, whether this protection occurs via the endothelial-derived H_2_S-mediated inhibition of the RhoA-ROCK_2_ pathway and its detailed mechanism still remain unclear. Therefore, the present study was designed to investigate the role of endothelial-derived H_2_S in inhibiting the RhoA-ROCK_2_ pathway for TFR-mediated protection against ischemic cerebral injury, focusing on the novel mechanism of regulating phosphorylation at the RhoA Ser188 and ROCK_2_ Thr436 residues.

## 2. Materials and Methods

### 2.1. Reagents and Drugs

The flowers of *Rhododendron simsii* Planch., authenticated by Prof. Shoujin Liu from the Department of Traditional Chinese Medicine Material Resources at Anhui University of Traditional Chinese Medicine (Hefei, China), were collected from the Dabie Mountains in Anhui Province, China. A voucher specimen (xH080843) has been deposited in the Herbarium of the Department of Pharmacology at Anhui Medical University. The extraction of TFR was carried out by Hefei Heyuan Medicine Technology Co., Ltd. (Hefei, China), with UV spectrophotometric analysis confirming a total flavonoid content exceeding 85%. The key pharmacoactive components in TFR were further characterized using ultra-performance liquid chromatography–tandem mass spectrometry (UPLC-MS/MS) via a Waters ACQUITY UPLC system integrated with an AB6500 Plus triple quadrupole mass spectrometer for precise compositional profiling. Sodium hydrosulfide (NaHS, catalog number: 161527) and DL-propargylglycine (PPG, catalog number: P7888) were purchased from Sigma Aldrich Chemical Co., Ltd. (St. Louis, MO, USA). Anti-ROCK_2_ antibodies (lot number: TD-M10-010), anti-p-ROCK_2_ (Thr436) antibodies (lot number: TD-M10-011), and anti-CSE antibodies (catalog number: BD-PN0625) were purchased from Suzhou Bo’aolong Technology Co., Ltd. (Suzhou, China). Anti-p-RhoA (Ser188) antibodies (catalog number: ab41435), anti-RhoA antibodies (catalog number: ab187027), and Cell Counting Kit-8 (CCK-8) (catalog number: ab228554) were purchased from Abcam (San Francisco, California, USA). Anti 3-MST antibodies (catalog number: sc-376168) were purchased from Santa Cruz Biotechnology (Santa Cruz, CA, USA). LDH test kit (catalog number: A020-2-2) was obtained from Nanjing Jiancheng Biotechnology Research Institute Co., Ltd. (Nanjing, China). RhoA activity (catalog number: MM-45317M2), ROCK_2_ activity (catalog number: MM-0886R2), H_2_S (catalog number: MM-21189R1), and NSE (catalog number: MM-0069R1) assay kits were obtained from Jiangsu Meimian Industrial, Co., Ltd. (Yancheng, China). CCG-1423 (catalog number: C5803) was purchased from ApexBio (Houston, TX, USA). KD-025 (catalog number: HY-15307) was purchased from MedChemExpress (Shanghai, China).

### 2.2. Animals

Neonatal Sprague Dawley (SD) rats born within 24 h and adult SD rats were bought from the Experimental Animal Center of Anhui Medical University. Wild-type (WT), ROCK_2_ heterozygote knockout (ROCK_2_ HK) and CSE knockout (CSE KO) mice, and WT and 3-MST knockout (3-MST KO) rats were obtained from Shanghai Biomodel Organism Science & Technology Development Co., Ltd. (Shanghai, China) All animals were housed in the Animal Center of Anhui Medical University, where they had unrestricted access to food and water. The environmental conditions for raising the animals were maintained at a humidity level of 54 ± 2% and a temperature of 22 ± 2 °C. The experiments received approval from the Ethical Committee of Anhui Medical University, and all experimental protocols adhered to the regulations established by the Animal Care and Use Committee at Anhui Medical University. These protocols comply with the guidelines outlined in the “*Guide for the Care and Use of Laboratory Animals*,” published by the US National Institutes of Health (NIH Publication No. 85-23, revised 2011).

The homozygous knockout of the ROCK_2_ gene is lethal; therefore, the experiment utilized heterozygotes. The CSE KO mouse model, widely used in H_2_S research, benefits from mature gene editing technology and well-characterized phenotypes [28,29]. Meanwhile, an existing study supports the application of 3-MST knockout rats in experimental research [30]. Given the species differences between rats and mice relevant to these three gene knockout models, this study employs two distinct experimental groups: one based on rats and the other on mice. These models have been clearly delineated and utilized separately in each experiment.

### 2.3. Isolation and Cultivation of ECs

The cerebrovascular ECs were isolated and cultured according to a previously established method [30] with some modifications. The rats or mice were anesthetized with pentobarbital sodium, decapitated, disinfected, and their cerebral cortices were placed in pre-cooled Petri dishes containing phosphate-buffered saline (PBS). The basilar artery, middle cerebral artery vessels, and other microscopic vessels were meticulously excised under a microscope. A total of 500 µL of PBS was added to a 1.5 mL centrifuge tube containing the freshly isolated blood vessel segment, which was subsequently cut into smaller pieces using ophthalmic scissors. Sterilized collagenase was introduced into the centrifuge tube along with the blood vessel segment before sealing it securely. The centrifuge tube was then placed in a water bath at 37 °C for digestion, with gentle shaking occurring every five min throughout this process. Following digestion, the mixture was subjected to centrifugation at 1400 r/min for ten min, then the supernatant was discarded and replaced with 1 mL of endothelial cell medium added to the pellet prior to transferring it into a culture bottle. After an incubation period of 24 h, non-adherent cells and tissue debris were removed by changing the medium, subsequently, cell morphology and growth were monitored every two to three days, with fluid changes performed as necessary. Once cell density reached approximately 80%, primary cultured cerebral vascular ECs were passed into a 6-well plate and identified through immunofluorescence techniques.

### 2.4. H/R Injury Model

When the cell culture reached the experimental requirements, after 24 h of pretreatment, the original medium was aspirated and replaced with a sugar-free medium. The cells were then cultured in a low-oxygen incubator (1% O_2_, 95% N_2_, 4% CO_2_, 37 °C) for 4 h. At the end of the incubation, the sugar-free medium was discarded and replaced with complete medium and placed in a normal incubator for 12 h for the next experiment.

WT mice cerebrovascular ECs were divided into the following groups: control group, H/R group, and 0.1 μmol/L [31] ROCK_2_ inhibitor KD-025 group.

WT and ROCK_2_ HK mice cerebrovascular ECs were, respectively, assigned to the following groups: control group and H/R group.

These two experiments were designed to investigate the effects of the ROCK_2_ inhibitor and ROCK_2_ HK on the protein expressions of CSE and 3-MST, as well as H_2_S production in cerebrovascular ECs subjected to H/R injury.

WT and CSE KO mice cerebrovascular ECs were, respectively, divided into four groups: control group, H/R group, 200 μmol/L [27] NaHS group, and 810 mg/L [7] TFR group.

WT and 3-MST KO rats cerebrovascular ECs were, respectively, divided into four groups: control group, H/R group, 200 μmol/L NaHS group, and 810 mg/L TFR group.

The two aforementioned experiments were designed to investigate the roles of CSE and 3-MST in TFR-promoted H_2_S production, as well as their association with the RhoA-ROCK_2_ signaling pathway in cerebrovascular ECs.

### 2.5. Cultivation of Hippocampal Neurons

The primary hippocampal neurons were cultured as previously described [32]. Several neonatal rats were captured within 24 h of birth. Following the administration of anesthesia, their skin was disinfected by immersion in 75% alcohol, and the brains were promptly excised through decapitation. The hippocampal tissues were meticulously excised in a pre-cooled Petri dish with PBS. A small glass bottle was prepared for autoclaving and drying, into which 0.125% trypsin solution was added. Subsequently, the removed hippocampal tissue was transferred to this small glass bottle with the 0.125% trypsin solution. The tissue was then cut using surgical scissors and allowed to rest on the table for 2 min. Following this, the glass bottle was placed in a cell incubator at 37 °C for digestion, which lasted between 15 and 20 min. During the digestion process, gentle shaking of the glass bottle occurred every 5 min to ensure thorough digestion. After completion of digestion, an equal volume of complete medium was added to terminate the enzymatic activity. The culture medium was gently aspirated using a pipette and set aside on the table. To remove insoluble matter, a nylon filter cloth with a mesh size of 200 mesh was employed; subsequently, centrifugation at room temperature at 1200 r/min for 5 min followed. Post centrifugation, the supernatant was discarded while complete medium was introduced back into the precipitate to re-suspend cells thoroughly by gentle mixing with a pipette gun before counting. Cell counts were adjusted to approximately 5 × 10^5^ cells/mL. The cell suspension was then seeded into pre-coated polylysine 6-well plates and cultured in a cell incubator maintained at 37 °C with an atmosphere of 5% CO_2_. After an incubation period of 24 h, media changes were performed using basic medium, thereafter, media changes continued every two days.

### 2.6. Co-Culture System

In this study, the Transwell co-culture system of hippocampal neurons and cerebral vascular ECs was established. The 6-well plate was chosen for the co-culture system, with a pet membrane featuring a pore size of 0.4 μm selected for the Transwell chamber. Primary hippocampal neurons were directly seeded onto the pore plate. After 7 days of cell growth, primary cultured cerebral vascular ECs were dissociated with pancreatic enzyme and then seeded in the upper chamber of the Transwell system placed on top of the pore plate, successfully establishing a co-culture system with cerebral vascular ECs in the upper chamber and hippocampal neurons in the lower chamber. The co-culture system was maintained in a 5% CO_2_ incubator at 37 °C for 24 h before subsequent experimental procedures.

Neurons+ECs^WT^ and Neurons+ECs^CSE KO^ co-cultured cells were divided into a control group and H/R group, respectively.

Neurons+ECs^WT^ and Neurons+ECs^3-MST KO^ co-cultured cells were divided into a control group and H/R group, respectively.

The above two experiments observed the effects of CSE and 3-MST KO cerebrovascular ECs on H_2_S content, H/R injury, and RhoA-ROCK_2_ pathway in the co-cultured neurons.

Neurons+ECs^WT^ and Neurons+ECs^CSE KO^ co-cultured cells were divided into a control group, H/R group, 200 μmol/L NaHS group, and 810 mg/L TFR group.

Neurons+ECs^WT^ and Neurons+ECs^3-MST KO^ co-cultured cells were divided into a control group, H/R group, 200 μmol/L NaHS group, and 810 mg/L TFR group.

Two types of H_2_S synthetases, CSE and 3-MST, are predominantly expressed in cerebrovascular ECs. In order to clarify the specific role of TFR in promoting H_2_S catalyzed by CSE and 3-MST in ECs in neuronal H/R injury, rat neurons were co-cultured with WT, CSE KO, and 3-MST KO ECs in the above two experiments. Then, NaHS and TFR were added into the ECs culture medium, and CBS inhibitor aminooxyacid and 3-MST inhibitor L-aspartic acid was added into the neurons culture medium to exclude the interference of H_2_S produced by neurons on experimental results.

### 2.7. Determination of Cell Viability and Biochemical Measurement

The viability of cells was detected at 450 nm using a CCK-8 assay kit. We tested the content of neuron-specific enolase (NSE), lactate dehydrogenase (LDH), and H_2_S according to assay kits and the manufacturer’s protocols. The activity of RhoA and ROCK_2_ were measured by enzyme-linked immunosorbent assay kits, and all procedures followed the manufacturer’s guidelines.

### 2.8. Western Blotting

Western blotting analysis was used to measure the protein expressions of CSE, 3-MST, RhoA, p-RhoA (Ser188), ROCK_2_, and p-ROCK_2_ (Thr436). The total proteins were separated using SDS-PAGE (Beyotime, Nanjing, China), transferred to polyvinylidene fluoride (PVDF) membranes (Millipore, Boston, MA, USA). The membranes were blocked with 5% skim milk in tris-buffered saline with Tween-20 (TBST) for 1 h, then shortly washed and hatched overnight at 4 °C with specific primary antibodies. After washing three times with TBST, the membranes were hatched with a secondary antibody in TBST solution for 1 h at 37 °C and then washed. The bands were visualized using an enhanced chemiluminescence kit (Thermo, Rockford, IL, USA). The intensity of immunoreactive bands was quantified with the use of image J software (Fiji distribution, NIH Image, Bethesda, MD, USA https://imagej.net/ij/ accessed on 9 June 2025). Glyceraldehyde-3-phosphate dehydrogenase (GAPDH) and β-actin were selected as reference proteins based on their molecular weights relative to the target protein.

### 2.9. Vessel Experiment

According to a previous method [33], after euthanizing the animals, the brain was promptly removed and the basilar artery of the mouse or rat brain was isolated. The basilar artery was subsequently prepared into a cerebrovascular segment measuring approximately 3 mm in length. Subsequently, the cerebrovascular segment was inserted into two tungsten wires and mounted on a wire myograph in 5 mL chambers (DMT, Aarhus, Denmark) containing physiological salt dissolution (PSS) at 37 °C and continuously aerated with a mixture of 95% O_2_ and 5% CO_2_. Controlled by the measurement system’s software, the pressure within the vascular ring was gradually increased from 10 mmHg to 60 mmHg over a period of 30 min. This pressure then stabilized for an additional hour, during which time the PSS fluid in the tank was replaced every 15 min. After incubation with various drugs for 30 min, a contraction was induced with 30 mmol/L KCl. Once vascular ring contraction stabilized, TFR or NaHS at the required concentration was added to induce vasodilation. The percentage of vasodilation is calculated according to the following formula: the percentage of vasodilation = (blood vessel diameter after adding medication—blood vessel diameter after adding contraction agent)/(blood vessel diameter before adding contraction agent—blood vessel diameter after adding contraction agent) × 100%.

The basal arteries of rats were divided into the following groups: vehicle group, (10, 30, 90, 270, and 810 mg/L) TFR group, 1.0 μmol/L [34] CCG-1423 + TFR group, and 0.1 μmol/L KD-025 + TFR group. This experimental design aimed to investigate the effects of pretreatment with RhoA inhibitor and ROCK_2_ inhibitor on TFR-induced vasodilation.

The basal arteries of WT and ROCK_2_ HK mice were divided into a vehicle group and (1, 3.16, 10, 31.62, 100, 316, and 1000 μmol/L) NaHS group. This experiment was designed to investigate the changes in H_2_S-mediated cerebral vessel dilation in ROCK_2_ HK mice.

### 2.10. Cerebral I/R Model

The cerebral I/R model was established by 20 min of bilateral common carotid artery occlusion (2-VO) three times with 10 min intervals, as previously described [35]. Briefly, the mice were anesthetized with 0.35% pentobarbital sodium (0.1 mL/10 g) via intraperitoneal injection. A midline incision was then made in the neck to expose the bilateral common carotid arteries (BCCAs). The BCCAs were isolated and fully ligated with sutures for a duration of 20 min to induce occlusion, followed by 10 min of reperfusion. This I/R protocol was repeated three times. Subsequently, the ligatures were carefully removed to restore blood flow. Finally, the midline incision in the neck was sutured closed.

WT mice were randomly divided into the following groups: sham group, I/R group, and 120 mg/kg [9] TFR group.

ROCK_2_ HK mice were randomly divided into the following groups: sham group, I/R group, 120 mg/kg TFR group, and 50 mg/kg [32] PPG + 120 mg/kg TFR group.

The mice in the TFR treatment groups were administrated with 120 mg/kg TFR by gavage once a day for 14 days. In addition to TFR treatment, mice of the PPG + TFR group were intraperitoneally injected daily with 50 mg/kg PPG for 14 days after cerebral I/R. The mice in cerebral I/R group were administrated with the same volume of normal saline. At 1 h after the first administration, the 2-VO-treated cerebral I/R injury was established. Mice in the sham group underwent exposure of BCCAs but not ligation. A heating station was used to maintain the rectal temperature of mice at 37.0 ± 0.5 °C during the operations. The experiment was designed to observe the effect of ROCK_2_ HK on TFR against ischemic cerebral injury and the effect of CSE inhibitor PPG treatment on TFR.

Detailed methods for the in vivo animal experiments are provided in Appendix A.

### 2.11. Determination of Blood Flow in Brain Tissue

The blood flow in mouse brain tissue was assessed using laser speckle flow imaging. Following anesthesia with pentobarbital sodium, the head hair was shaved and the skin and fascia were incised along the sagittal suture from the center of the head. The connective tissue on the skull surface was removed, and the body temperature was maintained at 37.0 ± 0.2 °C. The imaging system’s laser source emits a wavelength of 785 nm, uniformly irradiating the area approximately 9 cm above the mouse skull. In the real-time perfusion map generated by Pim Soft V1.4 software, detection regions for both the left and right hemispheres of each mouse brain were selected, allowing for the recording of cerebral blood flow (CBF) in both hemispheres. Using pre ischemic blood flow in mice as the baseline blood flow, the following calculation was used: change rate of CBF = reperfusion blood flow/pre ischemic blood flow × 100%.

### 2.12. Morris Water Maze Assay

The Morris water maze assay was performed as previously described [36] with some modifications. The mice were placed in the water maze for the observation of their swimming behavior, utilizing computer tracking. Each mouse’s swimming time was restricted to 60 s, with the duration from the onset of exploration until reaching the platform defined as the escape latency period. If a mouse failed to locate the platform within 60 s, it was gently guided to the platform and allowed a maximum stay of 10 s to observe spatial cues related to learning and memory. Subsequent training sessions over 3 days followed a clockwise direction, with an interval of at least 20 min between each session. Following 4 consecutive days of spatial learning, on day 5, the platform was removed for a spatial exploration test during which mice were permitted to swim freely for 60 s after being gently placed into the water from quadrant one.

### 2.13. Open Field Test

The open field test was performed as previously described [37]. The open field apparatus was a square box and the length, width, and height of it were, respectively, 96 cm, 96 cm, and 50 cm. Experimental mice were put in the central area of the open field equipment and permitted to move freely for 3 min to adapt to the environment. Afterwards, the exploratory behavior of the mice was then evaluated by recording the total distance traveled by the mice in an additional 3 min and the percentage of distance traveled by the mice within the boundary region.

### 2.14. Randomized Block Design

WT and gene knockout cells were, respectively, randomly allocated to the control group and the experimental group at an identical inoculation concentration. Synchronization was performed to minimize experimental error and potential confounding factors, such as genetic differences between species (e.g., rats and mice), thereby ensuring the robustness of the results. At the conclusion of each experiment, the effect of TFR was assessed by calculating the absolute difference between the study indicators obtained in the TFR group and the H/R group.

### 2.15. Statistical Analysis

All data, presented as mean ± SD, were analyzed using GraphPad Prism 8.0 software (La Jolla, CA, USA). A t-test was used to compare the differences between the two groups. One-way analysis of variance with Tukey’s test for post test analysis were used for multi-group comparisons. A value of *p* < 0.05 or *p* < 0.01 was considered statistically significant.

## 3. Results

### 3.1. ROCK_2_-Mediated Inhibition in CSE and 3-MST Expressions and H_2_S Production in Mouse Cerebrovascular ECs

The correlation between ROCK_2_ and H_2_S production in mouse cerebrovascular ECs was examined using the ROCK_2_ inhibitor as well as WT and ROCK_2_ HK mice. As shown in Figure 1A–C, compared with the control group, H/R injury led to significant reductions in CSE and 3-MST protein expression and H_2_S content in the cerebrovascular ECs from the WT mice, which were markedly reversed by the ROCK_2_ inhibitor KD-025 (0.1 μmol/L) compared with the H/R group, suggesting that the inhibition of ROCK_2_ could enhance CSE and 3-MST protein expression and H_2_S production in mouse cerebrovascular ECs subjected to H/R injury.

Figure 1D–F show that, in comparison with the control group, H/R injury resulted in significant reductions in CSE and 3-MST protein expression as well as H_2_S production in cerebrovascular ECs from the WT or ROCK_2_ HK mice, but the reductions in the ECs from the ROCK_2_ HK mice were remarkably attenuated compared with those in the ECs from the WT mice. The result suggested that the HK of ROCK_2_ could inhibit the reductions in CSE and 3-MST protein expression as well as H_2_S production in the ECs following H/R injury.

### 3.2. Effect of CSE or 3-MST KO on TFR Promoting H_2_S Production and Inhibiting the RhoA-ROCK_2_ Pathway in Mouse or Rat Cerebrovascular ECs Subjected to H/R Injury

Figure 1 clearly illustrates the inhibitory role of ROCK_2_ in H_2_S production within cerebrovascular ECs. To further explore this, we next investigated the regulatory effects of TFR-induced H_2_S on the RhoA-ROCK_2_ signaling pathway.

As shown in Figure 2A–E, H/R injury induced a significant decrease in H_2_S level and increases in RhoA and ROCK_2_ protein expression and activities in WT and CSE KO mouse cerebrovascular ECs and 3-MST KO rat cerebrovascular ECs, but these decreases and increases were markedly attenuated by treatment with TFR (810 mg/L) or H_2_S donor NaHS (200 μmol/L). The results demonstrated that, like NaHS, TFR could inhibit the upregulation of the RhoA-ROCK_2_ pathway in either WT or CSE KO or 3-MST KO ECs induced by H/R injury through H_2_S. The results of our paired experiment (Figure 2F) further showed that the TFR-caused increase in H_2_S level and decreases in protein expression and activities of RhoA and ROCK_2_ in the CSE KO or 3-MST KO ECs were substantially smaller than those in WT ECs. These results suggested that the KO of CSE or 3-MST attenuated the inhibitory effect of TFR on the H/R injury-induced decrease in H_2_S production and activation of the RhoA-ROCK_2_ pathway in mouse and rat cerebrovascular ECs.

### 3.3. Effects of CCG-1423 and KD-025 on the Cerebral Vasodilation of TFR

Vascular ECs regulate vascular tone by secreting various regulatory factors [38]. Accordingly, at the vascular level, we further examined the role of TFR and its relationship with both H_2_S and the RhoA-ROCK_2_ pathway.

As shown in Figure 3A, TFR 90, 270, and 810 mg/L induced a significant vasorelaxation in the rat cerebral basilar artery, which was remarkably attenuated by the RhoA inhibitor CCG-1423 (1.0 μmol/L) or ROCK_2_ inhibitor KD-025 (0.1 μmol/L), suggesting the involvement of the RhoA-ROCK_2_ signaling pathway in the TFR-induced cerebral vasorelaxation in rats. Additionally, Figure 3B shows that at the ranges of 10~1000 μmol/L, NaHS could elicit an obvious vasodilation in the cerebral basilar arteries from both WT and ROCK_2_ HK mice, but the vasodilation of NaHS was significantly more attenuated in the artery from ROCK_2_ HK mice than that from WT mice. This is consistent with our previous finding that endothelial-derived H_2_S induces rat cerebral vasodilation by inhibiting the RhoA-ROCK pathway [34], and suggested that the cerebral vasodilation of TFR may be due to the inhibition of the RhoA-ROCK_2_ signaling pathway by endothelial-derived H_2_S.

### 3.4. Protective Effects of TFR Against Cerebral I/R Injury in WT and ROCK_2_ HK Mice

The involvement of the RhoA-ROCK_2_ pathway in the cerebral vasorelaxation induced by TFR suggests that this pathway may also contribute to the protective effects of TFR against cerebral I/R injury. Thus, this study investigated the role of ROCK_2_ in TFR-mediated cerebral protection in mice.

As depicted in Figure 4A–C, cerebral I/R injury significantly induced a decrease in CBF and increases in serum LDH and NSE activities in WT and ROCK_2_ HK mice. The water maze test (Figure 4D) showed that the platform reaching latency increased but the platform crossing times decreased in both WT and ROCK_2_ HK mice subjected to cerebral I/R injury compared with the sham group. The open field test (Figure 4E) showed that cerebral I/R injury resulted in a significant decrease in moving distances and a marked increase in border area percentage in these two types of mice. Notably, the administration of TFR (120 mg/kg) significantly mitigated both the decreases and increases observed, suggesting that TFR provides a protective effect against cerebral I/R injury.

Figure 4 also shows that the cerebral I/R injury-induced decreases in CBF, spatial learning, memory functions, and exploratory behavior (such as platform reaching latency, platform crossing times, moving distances, and border area percentage) and increases in serum LDH and NSE activities were remarkably attenuated in ROCK_2_ HK mice compared with those in WT mice, and the amelioration of TFR on these indicators was markedly weaker in ROCK_2_ HK mice than that in WT mice. The results suggested that the HK of ROCK_2_ could reduce cerebral I/R injury, and ROCK_2_ was involved in the protection of TFR against brain I/R injury. Additionally, pretreatment with PPG (50 mg/kg), a CSE inhibitor, significantly attenuated the protection of TFR against cerebral I/R injury in ROCK_2_ HK mice, indicating that endogenous H_2_S might also participate in the protection of TFR in mice.

### 3.5. Effect of TFR on H/R Injury in the ECs-Co-Cultured Neurons and H_2_S Level in the Co-Cultured Medium

To observe the role of endothelial-derived H_2_S in H/R injury in neurons, the present study co-cultured rat neurons with WT and CSE KO (or 3-MST KO) cerebrovascular ECs, and then the H_2_S level in the co-cultured medium as well as neuronal vitality, LDH, and NSE activities were detected.

As shown in Figure 5A–C, during the co-culture of neurons with WT ECs, CSE KO ECs, or 3-MST KO ECs, H/R injury induced significant decreases in the neuronal vitality and the H_2_S level; at the same time, the activities of LDH and NSE in the neuronal supernatant were increased. Both NaHS (200 μmol/L) and TFR (810 mg/L) substantially inhibited the decreases in neuronal vitality and H_2_S level induced by H/R injury, as well as the increases in LDH and NSE activities observed in the neuronal supernatant during co-culture with WT ECs. Figure 5A,C shows that in the co-culture of neurons with CSE KO ECs, NaHS but not TFR significantly mitigated the reduction in neuronal vitality and H_2_S level, as well as the elevation of LDH and NSE activities in the neuronal supernatant induced by H/R injury. The results suggested that TFR could protect rat neurons from H/R injury through the CSE-produced H_2_S in the co-cultured ECs. Figure 5B,C shows that in the co-culture of neurons with 3-MST KO ECs, not only NaHS but also TFR significantly inhibited the H/R injury-decreased neuronal vitality and H_2_S level, and attenuated the increased LDH and NSE activities in the neuronal supernatant induced by H/R injury. Figure 5D further shows that the TFR-caused increases in decreased neuronal vitality and H_2_S level, as well as decreases in the increased LDH and NSE activities, were remarkably attenuated in the co-culture of neurons with 3-MST KO ECs compared with those in the co-culture of neurons with WT ECs. The results suggested that the 3-MST-produced H_2_S in the co-cultured ECs was also involved in the protection of TFR against H/R injury in rat neurons, but it seemed not as crucial as the CSE-produced H_2_S.

### 3.6. Impact of CSE or 3-MST KO in the ECs on the Effects of TFR on RhoA Expression, Activity, and Phosphorylation in the Co-Cultured Neurons Subjected to H/R Injury

As shown in Figure 6A–C, compared with the control group, the neurons co-cultured with WT ECs, CSE KO ECs, or 3-MST KO ECs exhibited significant increases in RhoA protein level and activity, along with a decrease in p-RhoA (RhoA phosphorylated at Ser188) protein level following H/R injury, which were remarkably inhibited by NaHS (200 μmol/L). In the neurons co-cultured with WT ECs, TFR (810 mg/L) had notable inhibitory effects on these increases and decreases. Figure 6A–C also demonstrates that, although there were diminished trends in TFR’s inhibition of the increased RhoA protein level and activity in the neurons co-cultured with CSE KO ECs or with 3-MST KO ECs, as well as reduced trends in TFR’s inhibition of the decreased p-RhoA protein level in the neurons co-cultured with 3-MST KO ECs, significant differences remained when compared with the H/R group. However, Figure 6A shows that TFR lost its inhibitory effect on the H/R injury-induced decrease in p-RhoA protein level in the neurons co-cultured with CSE KO ECs. Furthermore, Figure 6D,E indicate that the decreases in RhoA protein level and activity caused by TFR in the neurons co-cultured with either CSE KO ECs or 3-MST KO ECs, as well as the increase in decreased p-RhoA protein level observed in the neurons co-cultured with 3-MST KO ECs, were significantly attenuated compared with those seen in the neurons co-cultured with WT ECs. These findings indicate that TFR may inhibit the upregulation of RhoA expression and activation, as well as the downregulation of RhoA phosphorylation induced by H/R injury in rat neurons, and this effect appears to be mediated through CSE and 3-MST in co-cultured ECs.

### 3.7. Effects of CSE or 3-MST KO in the ECs on TFR Inhibiting the H/R Injury-Increased ROCK_2_ Expression, Activity, and Phosphorylation in the Co-Cultured Neurons

As shown in Figure 7A–C, H/R injury remarkably increased the ROCK_2_ and p-ROCK_2_ (ROCK_2_ phosphorylated at Thr436) protein levels and ROCK_2_ activity in the neurons co-cultured with WT ECs, CSE KO ECs, or 3-MST KO ECs compared with the control group. These increases were significantly inhibited by NaHS (200 μmol/L); except having no effect on the increased p-ROCK_2_ in the neurons co-cultured with CSE KO ECs, TFR (810 mg/L) notably attenuated the other increases. However, Figure 7D,E show that the TFR attenuated the H/R injury-increased ROCK_2_ and p-ROCK_2_ protein levels and ROCK_2_ activity, which were markedly weakened in the neurons co-cultured with CSE KO ECs or with 3-MST KO ECs compared with those in the neurons co-cultured with WT ECs. These results suggested that TFR could inhibit the H/R injury-increased ROCK_2_ expression, activity, and phosphorylation at the Thr436 site in rat neurons via the CSE and 3-MST in the co-cultured ECs.

### 3.8. Effects of Endothelial-Derived H_2_S on H/R Injury and the RhoA-ROCK_2_ Pathway in the Co-Cultured Neurons

To observe the role of the RhoA-ROCK_2_ pathway in the effect of H_2_S produced by CSE or 3-MST in cerebrovascular ECs against neuronal H/R injury, the present study co-cultured rat neurons with WT, CSE KO, and 3-MST KO cerebrovascular ECs from both mice and rats.

As shown in Figure 8, in the co-cultures of neurons with WT ECs, CSE KO ECs, and 3-MST KO ECs, the control groups did not show significant differences in H_2_S level and LDH and NSE activities in the culture medium, and no significant differences were observed in neuronal viability and RhoA activity and ROCK_2_ protein level in the co-cultured neurons. H/R exposure remarkably reduced the H_2_S level and neuronal viability, and increased the LDH, NSE, and RhoA activities and the ROCK_2_ protein level, suggesting that in the co-cultures, H/R exposure resulted in neuronal injury and the activation of the RhoA-ROCK_2_ pathway in the neurons, as well as a reduction in the H_2_S production.

Figure 8A–C further shows that the H/R exposure-induced decreases in the H_2_S level and neuronal viability and increases in the LDH and NSE activities were more significant in the co-culture of neurons with CSE KO ECs or 3-MST KO ECs than those in the co-culture of neurons with WT ECs, indicating that endothelial CSE and 3-MST mediated the H_2_S production in the co-cultured ECs during H/R injury and played a protective role against neuronal H/R injury. Figure 8D shows that during H/R injury, the RhoA activity and ROCK_2_ protein level also increased more significantly in the neurons co-cultured with CSE KO ECs or 3-MST KO ECs compared with those in the neurons co-cultured with WT ECs, suggesting that endothelial CSE and 3-MST exerted an inhibitory effect on the H/R injury-induced activation of the RhoA-ROCK_2_ pathway in co-cultured neurons.

## 4. Discussion

Ischemic brain injury includes neuronal and cerebral vascular damage, with cerebrovascular injury prior to neuronal lesion occurring [39,40]. Vascular ECs play a crucial role in the maintenance of cerebrovascular function. Injury to these cells leads to a decreased production of endothelial-derived relaxing factors, such as nitric oxide and H_2_S, resulting in impaired vasorelaxation, reduced CBF, and exacerbated ischemic brain injury [41]. A previous study showed that TFR had a protective effect against cerebral I/R injury, as indicated by the decrease in intracellular enzyme release from brain cells and the reduction in cerebral infarction [7]. Using a mouse model of cerebral I/R injury, the results of this study not only showed that TFR attenuated the releases of LDH and NSE from brain cells, but also, for the first time, reveal that TFR significantly mitigates spatial learning memory impairment and inquiry dysfunction induced by brain I/R injury in mice. Furthermore, the CSE inhibitor PPG markedly attenuates the protective effect of TFR on brain I/R injury in ROCK_2_ HK mice, suggesting that endogenous H_2_S is involved in the neuroprotective mechanism of TFR.

CBF is closely correlated to the occurrence and development of ischemic cerebral injury [42]. The present study indicated that TFR markedly increased the I/R injury-decreased CBF in WT and ROCK_2_ HK mice; this is beneficial to alleviate ischemic brain injury. Our vascular experiment further showed that TFR relaxed a rat cerebral artery, which is consistent with its increasing effect on the CBF. The RhoA-ROCK pathway is closely associated with cardiovascular and cerebrovascular physiological functions and pathological processes [43,44,45]. In the present study, the cerebral vasodilation of TFR in rats was markedly attenuated by the RhoA inhibitor CCG-1423 or ROCK_2_ inhibitor KD-025, suggesting an involvement of the RhoA-ROCK_2_ pathway in the TFR-induced cerebral vasorelaxation. In the cerebrovascular ECs of mice and rats, we observed that the KO of CSE or 3-MST substantially diminished TFR’s inhibitory effect on the reduction of H_2_S production and the activation of the RhoA-ROCK_2_ pathway induced by H/R injury. Combined with the result that PPG attenuated the increasing effect of TFR on the I/R injury-decreased CBF in ROCK_2_ HK mice, these data demonstrated that TFR promoted endothelial-derived H_2_S generation to inhibit the RhoA-ROCK_2_ pathway, thereby relaxing cerebral arteries and increasing CBF. This is one of the important mechanisms of the protective effect of TFR against ischemic brain injury.

The present study also observed that NaHS, an H_2_S donor, elicited an evident effect of inducing cerebral vasodilation in both WT and ROCK_2_ HK mice; however, this effect was significantly weaker in ROCK_2_ HK mice compared with WT mice. This is not only in agreement with a previous study that showed that endothelial-derived H_2_S can induce cerebral vasodilation by inhibiting the RhoA-ROCK pathway in rats [34], but also supports the above-mentioned finding that TFR relaxed cerebral arteries via promoting endothelial-derived H_2_S production to inhibit the RhoA-ROCK_2_ pathway.

There is a reciprocal inhibitory mechanism between H_2_S generation and the RhoA-ROCK pathway in the ECs, where endothelial-derived H_2_S inhibits the activation of the RhoA-ROCK pathway, and this pathway can in turn attenuate endothelial-derived H_2_S production [34]. This study demonstrated that the ROCK_2_ inhibitor KD-025 or the HK of ROCK_2_ significantly mitigated reductions in CSE and 3-MST protein expression, as well as H_2_S production in ECs following H/R injury, suggesting that the RhoA-ROCK_2_ pathway play a role in inhibiting the generation of H_2_S produced by endothelial CSE and 3-MST during H/R injury. This may contribute to the explanation for TFR relaxing the cerebral artery, as an increase in endothelial-derived H_2_S production induced by TFR mitigates the activation of the RhoA-ROCK_2_ pathway, which subsequently alleviates the inhibition on endothelial-derived H_2_S generation and enhances endothelium-dependent vasodilation in the cerebral arteries.

Endogenous H_2_S is not only an important endothelial-derived relaxing factor to act on blood vessels, but also a gaseous neurotransmitter for nerve cells. However, few studies have investigated the effects of endothelial-derived H_2_S on brain cells. In the co-culture of rat neurons with cerebrovascular ECs, our study showed that TFR had a significant protective effect against neuronal H/R injury; this is consistent with our previous study [46]. This study further suggests that the protective effect of TFR on neuronal H/R injury is mediated by H_2_S catalyzed by CSE and 3-MST in co-cultured ECs. It was also observed that both CSE- and 3-MST-catalyzed H_2_S play a role in the protective effect of TFR against neuronal H/R injury. However, following the KO of CSE in ECs, TFR no longer enhances H_2_S level in neurons or mitigates neuronal H/R damage. In contrast, after the KO of 3-MST, although the efficacy of TFR is diminished, it still increases H_2_S content in neurons and reduces neuronal damage from H/R. These findings suggest that TFR may primarily promote the production of H_2_S through CSE rather than via 3-MST within cerebrovascular ECs.

The overactivation of the RhoA-ROCK_2_ pathway in the brain leads to neuronal injury, whereas the inhibition of this pathway exerts a protective effect against cerebral I/R injury [47,48,49]. The phosphorylation of RhoA at Ser188 leads to an inactivation of itself and subsequent inhibitions on ROCK_2_ activity and expression, and H_2_S can protect rat neurons from H/R injury via promoting the phosphorylation of RhoA at Ser188 [24]. There are multiple phosphorylation sites in ROCK_2_, and phosphorylation at different sites does not obtain the same effects as ROCK_2_ activity. For instances, the phosphorylation at Tyr722 can inhibit ROCK_2_ activation [50], but the phosphorylation at Thr436 promotes the activation of ROCK_2_ and mediates the inhibitory effect of H_2_S on ROCK_2_ activity [27]. In the present study, the co-culture experiment indicated that TFR inhibited the H/R injury-induced increases in the RhoA and ROCK_2_ activities and p-ROCK_2_ (Thr436) protein level and decrease in the p-RhoA (Ser188) protein level in the neurons through the CSE and 3-MST in the co-cultured ECs. Together with the above-mentioned fact that endothelial-derived H_2_S is produced by CSE and 3-MST, these results demonstrated that TFR could inhibit the H/R injury-induced activation of the RhoA-ROCK_2_ pathway in the neurons through endothelial CSE- and 3-MST-produced H_2_S, which promotes the phosphorylation of RhoA at Ser188 and inhibits the phosphorylation of ROCK_2_ at Thr436, which aligns with our previous findings on the effects of exogenous H_2_S on the phosphorylation of RhoA at Ser188 and ROCK_2_ at Thr436 [24,27]. Furthermore, the present study also indicated that via the CSE and 3-MST in the co-cultured ECs, TFR attenuated the neuronal RhoA and ROCK_2_ protein levels increased by H/R injury. The results similarly indicated that TFR could inhibit the H/R injury-increased expression of the RhoA-ROCK_2_ pathway in the neurons through endothelial-derived H_2_S. Additionally, in the neurons and ECs co-culture systems, endothelial CSE and 3-MST were responsible for the production of H_2_S, which protected neurons from H/R injury by suppressing the H/R injury-induced upregulation of the RhoA-ROCK_2_ pathway. These findings collectively support the conclusion that TFR mitigates H/R injury-induced neuronal damage through the endothelial-derived H_2_S-mediated inhibition of the RhoA-ROCK_2_ signaling pathway.

While consistent with established mechanisms of the neuroprotective effects of neuronal CBS-derived H_2_S against ischemic brain injury [8] and exogenous H_2_S-mediated ROCK_2_ inhibition [27,50], our study innovatively identifies endothelial-derived H_2_S as the critical paracrine mediator regulating neuronal RhoA-ROCK_2_ signaling in a clinically relevant co-culture model. Nevertheless, the phosphorylase activity and other potential phosphorylation sites (e.g., the Tyr722 site) involved in ROCK_2_ have not been fully explored in this study. Additionally, although homozygous ROCK_2_ knockout results in embryonic lethality, the heterozygous model retains functional ROCK_2_ (~50% expression), which may obscure the full phenotypic consequences. These limitations will be addressed in our future investigations.

## 5. Conclusions

In conclusion, the present study demonstrated that TFR can suppress the RhoA-ROCK_2_ pathway by enhancing the production of H_2_S catalyzed by CSE and 3-MST in cerebrovascular ECs, thereby contributing to cerebrovascular relaxation and alleviating brain I/R injury. In addition, the endothelial-derived H_2_S promoted by TFR may be mainly catalyzed by CSE. TFR exerts a neuroprotective effect by promoting endothelial-derived H_2_S, which inhibits the activation and expression of the RhoA-ROCK_2_ pathway induced by H/R injury in neurons. The inhibitory effect of TFR on this pathway activation is at least partially mediated by endothelial-derived H_2_S facilitating the phosphorylation of RhoA at Ser188 while concurrently inhibiting the phosphorylation of ROCK_2_ at Thr436 (Figure 9). These findings are significant for elucidating the regulatory mechanism of the H_2_S–RhoA-ROCK_2_ pathway in TFR’s anti-ischemic brain injury effect and provide valuable reference for subsequent clinical investigations on TFR.

## Figures and Tables

**Figure 1 cimb-47-00513-f001:**
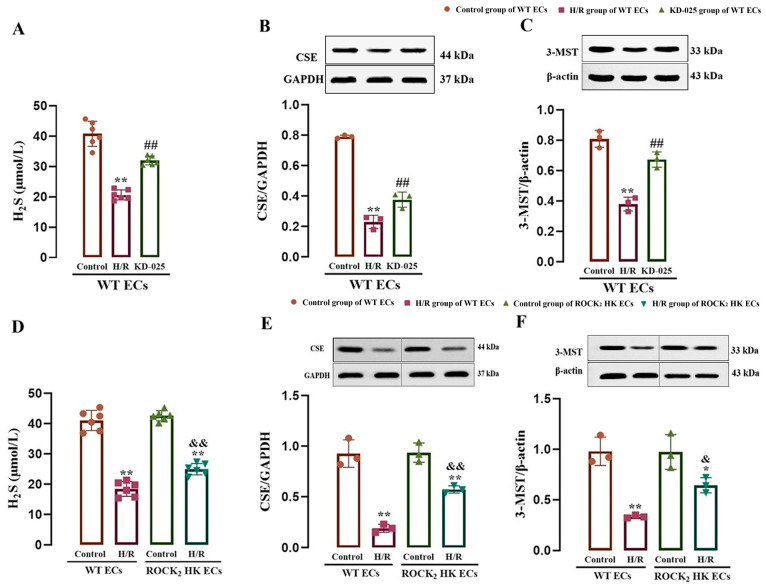
Effects of ROCK_2_ inhibitor KD-025 and HK of ROCK_2_ on H_2_S production and CSE and 3-MST protein expression in the mouse cerebrovascular ECs subjected to H/R injury (mean ± SD, *n* = 6 or 3). (**A**–**C**) Effect of KD-025 on the H_2_S content and CSE and 3-MST protein expression in the ECs. (**D**–**F**) Effect of HK of ROCK_2_ on the H_2_S content and CSE and 3-MST protein expression in the ECs. KD-025: 0.1 μmol/L. * *p* < 0.05, ** *p* < 0.01 vs. the control group; ^##^
*p* < 0.01 vs. the H/R group; ^&^
*p* < 0.05, ^&&^
*p* < 0.01 vs. the H/R group of WT ECs.

**Figure 2 cimb-47-00513-f002:**
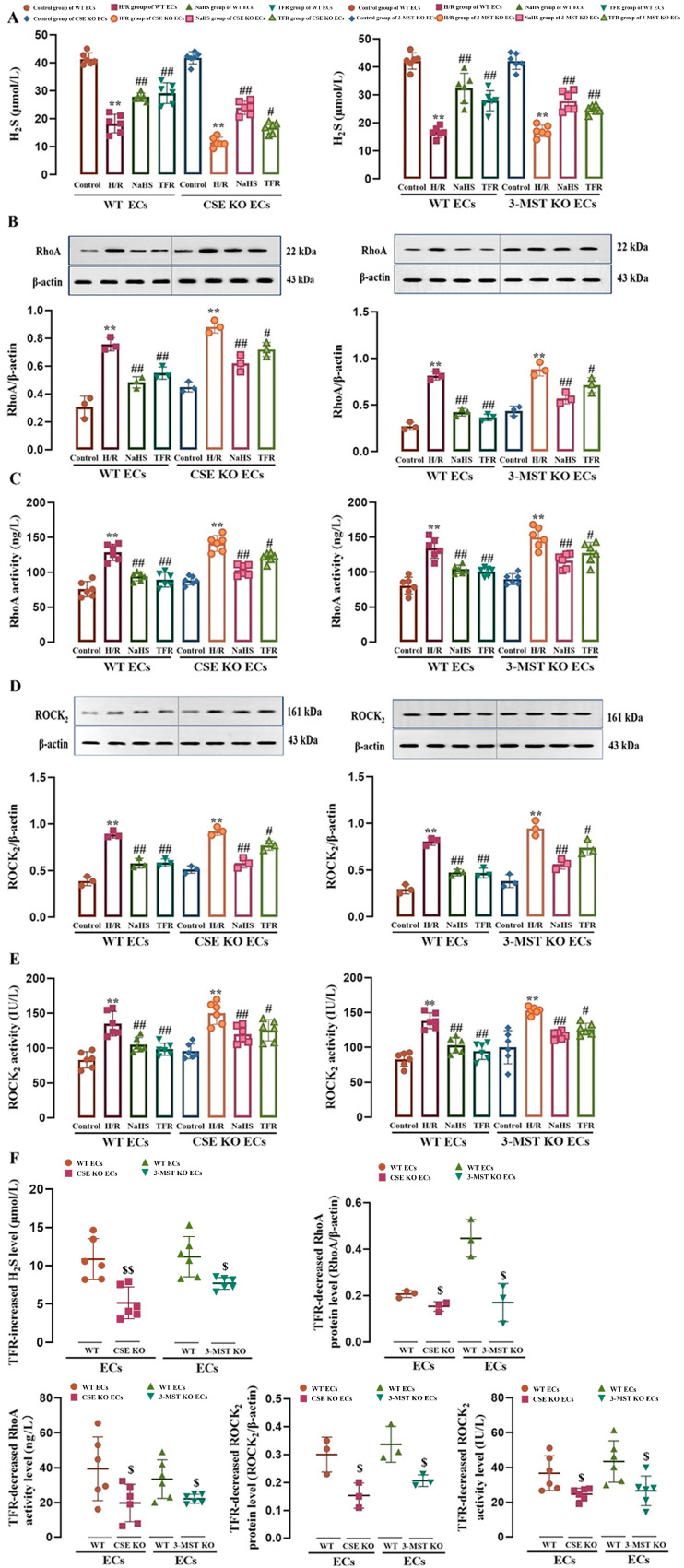
Effects of TFR and NaHS on the H_2_S level as well as RhoA and ROCK_2_ protein expression and activities in WT and CSE KO mouse cerebrovascular ECs (or WT and 3-MST KO rat cerebrovascular ECs) subjected to H/R injury (mean ± SD, *n* = 6 or 3). (**A**) H_2_S level. (**B**) RhoA protein level. (**C**) RhoA activity. (**D**) ROCK_2_ protein level. (**E**) ROCK_2_ activity. (**F**) Comparison of the TFR-induced increase in H_2_S level and decreases in protein expression and activities of RhoA and ROCK_2_ in WT and CSE KO ECs (or WT and 3-MST KO ECs). NaHS: 200 μmol/L; TFR: 810 mg/L. ** *p* < 0.01 vs. the control group; ^#^
*p* < 0.05, ^##^
*p* < 0.01 vs. the H/R group; ^$^
*p* < 0.05, ^$$^
*p* < 0.01 vs. the WT ECs group.

**Figure 3 cimb-47-00513-f003:**
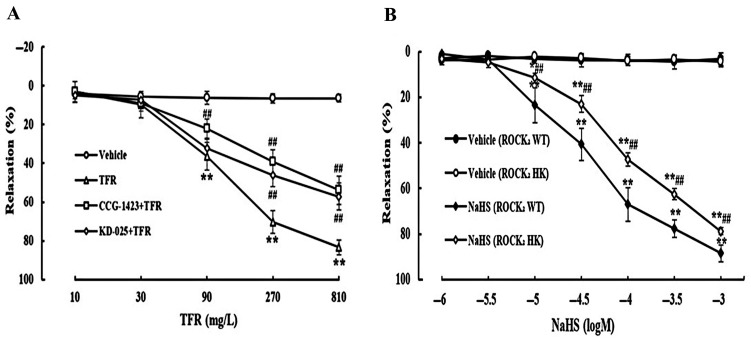
Effects of RhoA inhibitor CCG-1423 and ROCK_2_ inhibitor KD-025 on vasodilation of TFR in rat cerebral basilar artery, and impact of HK of ROCK_2_ on the NaHS-induced cerebral vasorelaxation in mice (mean ± SD, *n* = 6). (**A**) Effects of CCG-1423 (1.0 μmol/L) or KD-025 (0.1 μmol/L) pretreatment on cerebral vasodilation of TFR in rats. (**B**) Effect of HK of ROCK_2_ on the NaHS-induced cerebral vasodilation in mice. * *p* < 0.05, ** *p* < 0.01 vs. the vehicle group; ^##^
*p* < 0.01 vs. the TFR group or the NaHS (ROCK_2_ WT) group.

**Figure 4 cimb-47-00513-f004:**
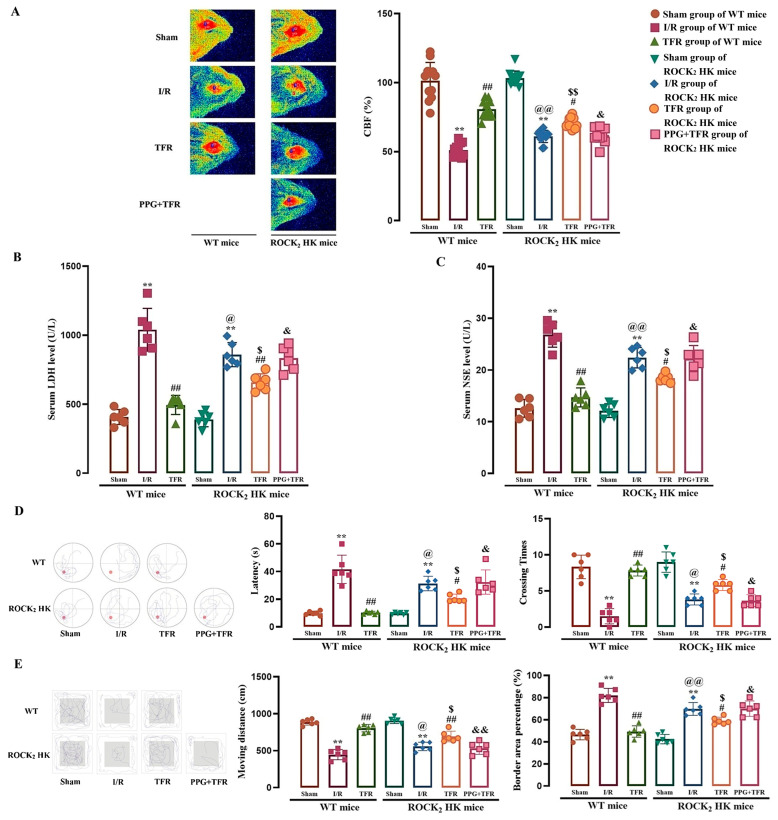
Protective effect of TFR against brain I/R injury in WT and ROCK_2_ HK mice, and impact of PPG on the protection of TFR in ROCK_2_ HK mice (mean ± SD, *n* = 6). (**A**) CBF (Red areas represent relatively normal blood perfusion, while blue areas represent hypoperfusion, one detection area per left/right cerebral hemisphere in each mouse). (**B**) Serum LDH level. (**C**) Serum NSE level. (**D**) Water maze test. (**E**) Open field test. TFR: 120 mg/kg, PPG: 50 mg/kg. ** *p* < 0.01 vs. the sham group; ^#^
*p* < 0.05, ^##^
*p* < 0.01 vs. the I/R group; ^@^
*p* < 0.05, ^@@^
*p* < 0.01 vs. the I/R group of WT mice; ^$^
*p* < 0.05, ^$$^
*p* < 0.01 vs. the TFR group of WT mice; ^&^
*p* < 0.05, ^&&^
*p* < 0.01 vs. the TFR group of ROCK_2_ HK mice.

**Figure 5 cimb-47-00513-f005:**
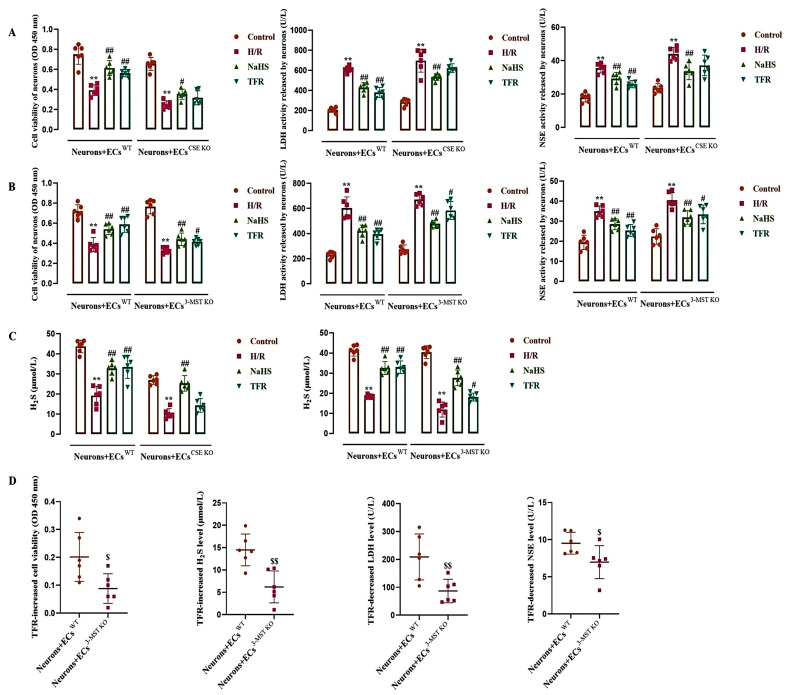
Effects of TFR and NaHS on H/R injury in the rat neurons co-cultured with cerebrovascular ECs and H_2_S level in the co-cultured medium (mean ± SD, *n* = 6). (**A**) Effects of TFR and NaHS against H/R injury in the rat neurons co-cultured with WT and CSE KO ECs. (**B**) Effects of TFR and NaHS against H/R injury in the rat neurons co-cultured with WT and 3-MST KO ECs. (**C**) Effects of TFR and NaHS on the H_2_S level in the co-cultured medium. (**D**) Comparison of the TFR-induced increases in neuronal vitality and H_2_S level, along with the decreases in LDH and NSE activities, between the co-culture of neurons with WT ECs and that with 3-MST KO ECs. NaHS: 200 μmol/L; TFR: 810 mg/L. ** *p* < 0.01 vs. the control group; ^#^
*p* < 0.05, ^##^
*p* < 0.01 vs. the H/R group; ^$^
*p* < 0.05, ^$$^
*p* < 0.01 vs. the Neurons+ECs^WT^ group.

**Figure 6 cimb-47-00513-f006:**
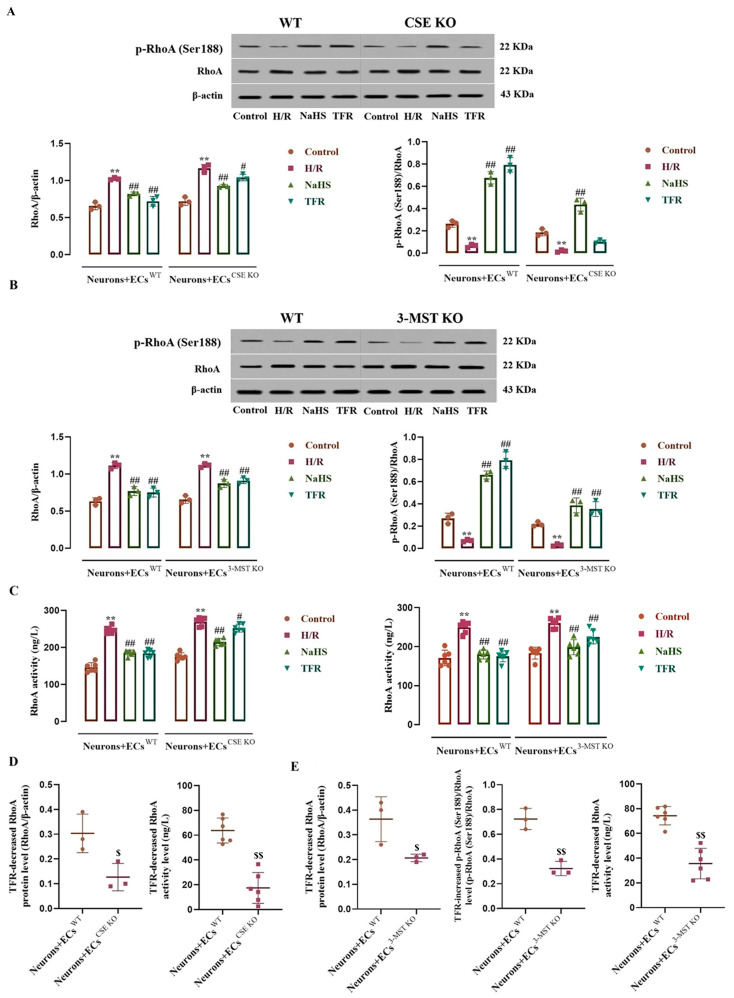
Effects of TFR and NaHS on the RhoA protein expression, activity, and phosphorylation in the rat neurons co-cultured with cerebrovascular ECs subjected to H/R injury (mean ± SD, *n* = 3 or 6). (**A**) RhoA and p-RhoA (RhoA phosphorylated at Ser188) protein levels in the neurons co-cultured with WT ECs or CSE KO ECs. (**B**) RhoA and p-RhoA protein levels in the neurons co-cultured with WT ECs or 3-MST KO ECs. (**C**) RhoA activity in the neurons co-cultured with WT ECs, CSE KO ECs, and 3-MST KO ECs. (**D**) Comparison of the TFR-caused decreases in the H/R injury-increased RhoA protein level and activity in the neurons co-cultured with WT ECs and CSE KO ECs. (**E**) Comparison of the TFR-caused decreases in the H/R injury-increased RhoA protein level and activity and increase in the decreased p-RhoA protein level in the neurons co-cultured with WT ECs and 3-MST KO ECs. NaHS: 200 μmol/L; TFR: 810 mg/L. ** *p* < 0.01 vs. the control group; ^#^
*p* < 0.05, ^##^
*p* < 0.01 vs. the H/R group; ^$^
*p* < 0.05, ^$$^
*p* < 0.01 vs. the Neurons+ECs^WT^ group.

**Figure 7 cimb-47-00513-f007:**
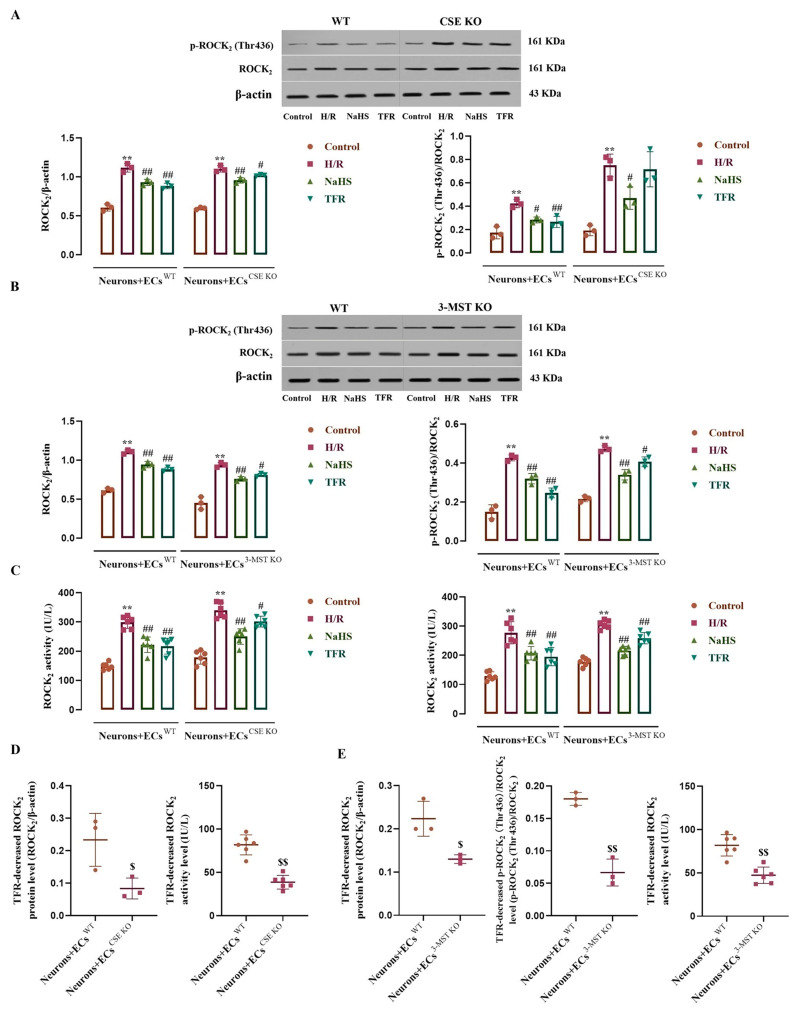
Effects of TFR and NaHS on the H/R injury-increased ROCK_2_ protein expression, activity, and phosphorylation in the rat neurons co-cultured with cerebrovascular ECs (mean ± SD, *n* = 3 or 6). (**A**) ROCK_2_ and p-ROCK_2_ (ROCK_2_ phosphorylated at Thr436) protein levels in the neurons co-cultured with WT ECs or CSE KO ECs. (**B**) ROCK_2_ and p-ROCK_2_ protein levels in the neurons co-cultured with WT ECs or 3-MST KO ECs. (**C**) ROCK_2_ activity in the neurons co-cultured with WT ECs, CSE KO ECs, and 3-MST KO ECs. (**D**) Comparison of the TFR-decreased and the H/R injury-increased ROCK_2_ protein level and activity in the neurons co-cultured with WT ECs and CSE KO ECs. (**E**) Comparison of the TFR-decreased and the H/R injury-increased ROCK_2_ and p-ROCK_2_ protein levels and ROCK_2_ activity in the neurons co-cultured with WT ECs and 3-MST KO ECs. NaHS: 200 μmol/L; TFR: 810 mg/L. ** *p* < 0.01 vs. the control group; ^#^
*p* < 0.05, ^##^
*p* < 0.01 vs. the H/R group; ^$^
*p* < 0.05, ^$$^
*p* < 0.01 vs. the Neurons+ECs^WT^ group.

**Figure 8 cimb-47-00513-f008:**
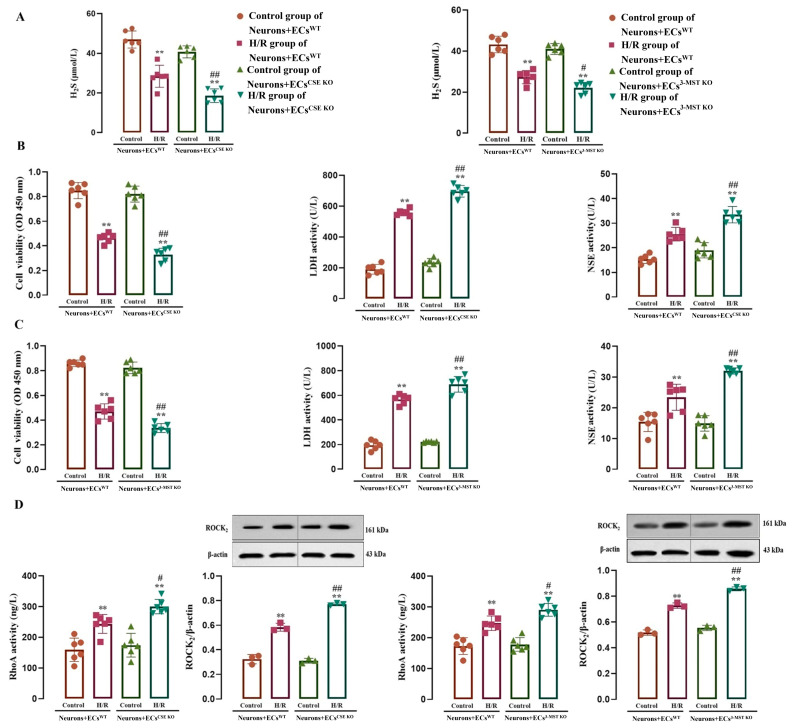
Effect of the H_2_S produced by CSE or 3-MST in mouse or rat cerebrovascular ECs on H/R injury, RhoA activity, and ROCK_2_ protein expression in rat neurons co-cultured with the ECs (mean ± SD, *n* = 6 or 3). (**A**) H_2_S level in the co-cultured medium. (**B**) H/R injury in the neurons co-cultured with the WT ECs or CSE KO ECs. (**C**) H/R injury in the neurons co-cultured with the WT ECs or 3-MST KO ECs. (**D**) RhoA activity and ROCK_2_ protein level in neurons co-cultured with the WT ECs, CSE KO ECs, and 3-MST KO ECs. ** *p* < 0.01 vs. the control group; ^#^
*p* < 0.05, ^##^
*p* < 0.01 vs. the H/R group in co-culture of Neurons+ECs^WT^.

**Figure 9 cimb-47-00513-f009:**
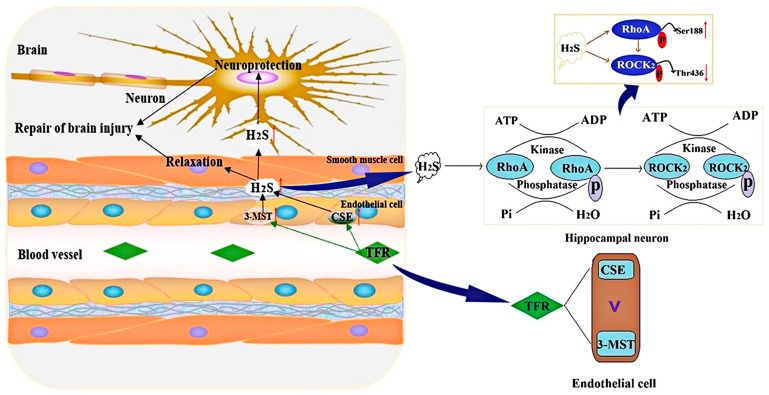
Schematic representation of the protective mechanism of TFR against ischemic cerebral injury. By enhancing the production of H_2_S catalyzed by CSE and 3-MST in cerebrovascular ECs, TFR can promote the phosphorylation of RhoA at Ser188 while inhibiting the phosphorylation of ROCK_2_ at Thr436. This mechanism contributes to cerebrovascular relaxation and alleviates cerebral I/R injury.

## Data Availability

The datasets utilized and/or analyzed in the current study are available from the corresponding author(s) upon reasonable request.

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
