# Peer review of "Total Flavones of Rhododendron Protect Against Ischemic Cerebral Injury by Regulating the Phosphorylation of the RhoA-ROCK2 Pathway via Endothelial-Derived H2S"

_cimb, 2025, doi:10.3390/cimb47070513_

Round 1

Reviewer 1 Report

Comments and Suggestions for Authors
  1. Both the introduction and the conclusion should discuss the innovation (e.g., differentiation from existing H2S or ROCK2 inhibitor studies). Additionally, the discussion section lacks sufficient discussion on the limitations of the study; therefore, it is advisable to provide additional details and enhancements.
  2. Why is the bilateral carotid artery occlusion (2-VO) model of global ischemia used instead of the middle cerebral artery occlusion (MCAO) focal ischemia model
  3. CSE/3-MST KO verification: Provide Western blot or activity assays to confirm protein knockout (currently relies only on genotyping).
  4. Given homozygous KO lethality, if residual ROCK2 activity in heterozygotes may influence results,
  5. Figures 6–7 show TFR regulates p-RhoA(Ser188) and p-ROCK2(Thr436), what about other potential sites (e.g., ROCK2-Tyr722)? Are they affected?
  6. ​Replace bar graphs in Figures 4–5 with scatter plots (individual data points + mean) for transparency.
  7. Compare findings more thoroughly with the authors’ prior work on H2S-ROCK2 (e.g., Ref 28).

​

Author Response

Reviewer 1

Comment 1: Both the introduction and the conclusion should discuss the innovation (e.g., differentiation from existing H2S or ROCK2 inhibitor studies). Additionally, the discussion section lacks sufficient discussion on the limitations of the study; therefore, it is advisable to provide additional details and enhancements.

Response 1: We sincerely thank the reviewer for the valuable suggestions. In the revised Introduction section, we have added content highlighting the innovative aspect of our study (page 2, paragraph 5, lines 90-94). In the revised Discussion section, we have supplemented: 1) comparisons with previous studies to highlight similarities and differences, and 2) an expanded discussion on study limitations (page 20, paragraph 3, lines 684-691), (page 21, paragraph 1, lines 692-693).

Comment 2: Why is the bilateral carotid artery occlusion (2-VO) model of global ischemia used instead of the middle cerebral artery occlusion (MCAO) focal ischemia model?

Response 2: Thank you for pointing this out. Our study focuses on investigating the ‌neuroprotective effects of the drug on hippocampal neuronal function‌. Given that the hippocampus is a critical brain region mediating learning and memory, the classic ‌middle cerebral artery occlusion (MCAO)‌ model was deemed unsuitable for simulating ‌chronic cognitive impairment‌ due to its inherent limitations: short post-ischemic survival time and motor dysfunction that confounds behavioral assessments. Consequently, we adopted the bilateral common carotid artery occlusion (2-VO) model, which induces hippocampal neuronal damage through ‌chronic cerebral hypoperfusion‌, thereby better recapitulating the pathological mechanisms underlying ‌vascular dementia‌ or ‌chronic cerebral ischemia-associated cognitive deficits‌. To quantify the drug’s impact on learning and memory, we employed the Morris water maze (to assess spatial memory) and open field test ‌(to assess exploratory behavior capability)‌.

Comment 3: CSE/3-MST KO verification: Provide Western blot or activity assays to confirm protein knockout (currently relies only on genotyping).

Response 3: We sincerely appreciate the reviewer’s insightful suggestion regarding protein-level validation for the CSE/3-MST knockout models. While Western blotting or activity assays would offer complementary protein-level data, our genotyping strategy has been rigorously validated in prior studies (References: 28-30), demonstrating its reliability for confirming knockout status. To further strengthen methodology transparency, we have added the following statement to the Materials and Methods section: " The CSE KO mouse model, widely used in Hâ‚‚S research, benefits from mature gene-editing technology and well-characterized phenotypes [28,29]. ‌Meanwhile‌, existing study support the application of 3-MST knockout rats in experimental research [30] (page 3, paragraph 3, lines 139-142)".

Comment 4: Given homozygous KO lethality, if residual ROCK2 activity in heterozygotes may influence results?

Response 4: We thank the reviewer for raising this important point. As noted in the revised ‌Discussion section (page 20, paragraph 3, lines 690-691), (page 21, paragraph 1, lines 692-693)‌, we have explicitly addressed the potential influence of residual ROCK2 activity in heterozygous models on our results.

Comment 5: Figures 6–7 show TFR regulates p-RhoA (Ser188) and p-ROCK2 (Thr436), what about other potential sites (e.g., ROCK2-Tyr722)? Are they affected?

Response 5: We sincerely thank the reviewer for highlighting this important point concerning additional phosphorylation sites, particularly ROCK2 Tyr722. We have explicitly addressed this gap in the revised Discussion section (page 20, paragraph 3, lines 688–690). Future studies will specifically investigate TFR's effects on ROCK2 Tyr722 phosphorylation and other potential regulatory sites to fully characterize the signalling mechanisms.​

Comment 6: Replace bar graphs in Figures 4–5 with scatter plots (individual data points + mean) for transparency.

Response 6: We appreciate the reviewer's comment and would like to clarify that Figures 4–5 already incorporate scatter plots overlaid on bar graphs, which display all individual data points while clearly indicating group means. We believe this presentation format meets the journal's data transparency requirements.

Comment 7: Compare findings more thoroughly with the authors’ prior work on H2S-ROCK2 (e.g., Ref 28).

Response 7: We thank the reviewer for highlighting the importance of contextualizing our findings with prior research. In the revised Discussion (page 20, paragraph 2, lines 672–674), we have expanded our comparative analysis with Reference 28 (now revised to Reference 27), which examined ‌exogenous Hâ‚‚S effects on ROCK2 Thr436 phosphorylation‌. Our current work specifically investigates how ‌TFR-induced endogenous Hâ‚‚S generation‌ regulates phosphorylation at this same site (Thr436), establishing a novel physiological complement to previous pharmacological findings.

Reviewer 2 Report

Comments and Suggestions for Authors

This research by Sun et al. investigated the protective effects of total flavones of Rhododendron on ischemic cerebral injury and the underlying mechanisms. The manuscript is well organized. The experimental work is comprehensive, with both in vivo and in vitro studies systematically confirming and elucidating the protective effects and mechanisms of TFR. Given the significant impact of ischemic brain injury on human health, this work is expected to attract broad interest among researchers in the fields of medicine and biology. Therefore, I highly recommend the publication of this manuscript in Curr. Issues Mol. Biol. Only minor revisions are needed to potentially improve the quality of the manuscript as follows:

  1. It is recommended that the authors improve the presentation of Figure 9 to more clearly indicate whether the signaling pathways represent promotion or inhibition.
  2. Sections 2.4 and 2.5 serve as good examples; it is recommended that the authors begin other sections by clarifying the experimental rationale and objectives rather than directly presenting the results.
  3. The authors conducted systematic experiments using various in vivo and in vitro models. Including schematic diagrams of the experimental workflows in the corresponding figures could enhance clarity, although this is not mandatory.
  4. The text in Figures 4D and 4E should be enlarged to improve readability.

The authors are advised to carefully check the manuscript for minor errors, such as:

  1. The order of Figures 2 and 3 is reversed; it is recommended to correct this.
  2. The symbol "S" used to indicate significance in the figure should be consistent with the symbol "$" referenced in the figure captain.
  3. Panel A in Figure 6 is incomplete and should be corrected for full display.

Author Response

Reviewer: 2

Comment 1: It is recommended that the authors improve the presentation of Figure 9 to more clearly indicate whether the signaling pathways represent promotion or inhibition.

Response 1: We thank the reviewer for this valuable suggestion. To enhance the clarity of signaling pathway interactions in Figure 9, we have comprehensively revised the figure (page 21, paragraph 3, line 707, Figure 9).

Comment 2: Sections 2.4 and 2.5 serve as good examples; it is recommended that the authors begin other sections by clarifying the experimental rationale and objectives rather than directly presenting the results.

Response 2: As recommended, we have revised ‌other sections‌ accordingly to clarify the experimental rationale and objectives upfront, mirroring the structure of Sections 2.4 and 2.5. These revisions are implemented in: (page 9, paragraph 2, lines 376–378),(page 9, paragraph 4, lines 393–395).

Comment 3: The authors conducted systematic experiments using various in vivo and in vitro models. Including schematic diagrams of the experimental workflows in the corresponding figures could enhance clarity, although this is not mandatory.

Response 3: We appreciate your suggestion and agree that schematic diagrams could enhance clarity. However, we have chosen to maintain the current figures to emphasize data presentation, as the key methodological details are already comprehensively described in the Methods section.

Comment 4: The text in Figures 4D and 4E should be enlarged to improve readability.

Response 4: We thank the reviewer for this constructive suggestion. The text in Figures 4D and 4E has been enlarged accordingly to improve readability in the revised manuscript (page 13, paragraph 2, line 506, Figure 4).

Comment 5: The order of Figures 2 and 3 is reversed; it is recommended to correct this.

Response 5: Thank you for your careful review and helpful suggestion. We have corrected the order of Figures 2 and 3 in the revised manuscript (page 10, paragraph 1, line 418, Figure 2),(page 11, paragraph 2, lines 427, Figure 3).

Comment 6: The symbol "S" used to indicate significance in the figure should be consistent with the symbol "$" referenced in the figure captain.

Response 6: Thanks for noting this. We've enlarged all '$' significance markers in figures for clarity.

Comment 7: Panel A in Figure 6 is incomplete and should be corrected for full display.

Response 7: Thank you for your feedback. Panel A in Figure 6 has been corrected for full display (page 15, paragraph 1, line521, Figure 6).